# Employment Industry and Occupational Class in Relation to Serious Psychological Distress in the United States

**DOI:** 10.3390/ijerph19148376

**Published:** 2022-07-08

**Authors:** Lauren R. Gullett, Dana M. Alhasan, W. Braxton Jackson, Chandra L. Jackson

**Affiliations:** 1Epidemiology Branch, National Institute of Environmental Health Sciences, National Institutes of Health, Department of Health and Human Services, Research Triangle Park, Durham, NC 27709, USA; lauren.gullett@nih.gov (L.R.G.); dana.alhasan@nih.gov (D.M.A.); 2Social & Scientific Systems, Inc., A DLH Holdings Company, Durham, NC 27703, USA; braxton.jackson@dlhcorp.com; 3Intramural Program, National Institute on Minority Health and Health Disparities, National Institutes of Health, Department of Health and Human Services, Bethesda, MD 20892, USA

**Keywords:** occupations, occupational health, psychological distress, employment, race factors, economic status, mental health

## Abstract

Occupational characteristics may influence serious psychological distress (SPD) and contribute to health inequities; yet, few studies have examined multiple employment industries and occupational classes in a large, racially diverse sample of the United States. Using data from the National Health Interview Survey, we investigated employment industry and occupational class in relation to SPD in the overall population and by race/ethnicity, gender, age, household income, and health status. We created eight employment industry categories: professional/administrative/management, agricultural/manufacturing/construction, retail trade, finance/information/real estate, educational services, health care/social assistance, accommodation/food services, and public administration/arts/other services. We also created three occupational class categories: professional/management, support services, and laborers. SPD was measured using the Kessler Psychological Distress Scale and scores ≥13 indicated SPD. We adjusted for confounders and used Poisson regression to estimate prevalence ratios (PRs) and 95% confidence intervals (CIs). Among the 245,038 participants, the mean age was 41.7 ± 0.1 years, 73% were Non-Hispanic (NH)-White, and 1.5% were categorized as having SPD. Compared to the professional/administrative/management industry, working in other industries (e.g., manufacturing/construction (PR = 0.82 [95% CI: 0.70–0.95]) and educational services (PR = 0.79 [95% CI: 0.66–0.94])) was associated with lower SPD. Working in support services and laborer versus professional/management positions were both associated with 19% higher prevalence of SPD (95% CI: 1.04–1.35; 95% CI: 1.04–1.38, respectively). Furthermore, working in a support services or laborer versus professional/management position was associated with higher SPD in most employment industries. Industry-specific workplace interventions to equitably improve mental health are warranted.

## 1. Introduction

Serious psychological distress (SPD) is a general measure of mental illness characterized by depressive symptoms and anxiety [1]. SPD interferes with one’s ability to engage in activities of daily life and is estimated to directly affect 10 million adults in the United States [2]. Associated with many chronic and life-threatening conditions, including diabetes, chronic obstructive pulmonary disease, and cardiovascular disease [3,4,5,6], a disproportionate burden of SPD has been experienced by non-Hispanic (NH)-Black and Hispanic/Latinx individuals, women, middle-aged and older adults, individuals with low-income, and adults with chronic conditions (e.g., physical disabilities) [2,7]. Therefore, SPD is an important public health problem.

A growing body of research has examined the impact of occupational characteristics (e.g., shiftwork among nurses, working as a woman in a male-dominated industry) on SPD and, more generally, poor mental health [8,9,10]. Specifically, working in certain occupational classes (e.g., support service and laborious roles such as nursing assistants) and high-intensity employment industries (e.g., agriculture, healthcare, and education) with high stress levels may be associated with SPD through various socioenvironmental pathways that have physiological consequences [11]. In support service or laborious positions, employees may have demanding workloads with low decision-making latitude (e.g., job strain), work long hours and/or have nonstandard schedules, receive insufficient compensation or benefits to support a healthy lifestyle, and may be subjected to harmful and oppressive working conditions, which can lead to demoralization and stress [10,11,12,13,14,15]. Similarly, adults who work in high-intensity employment industries often receive insufficient compensation and may experience work-related physical and emotional exhaustion [10,16,17]. This may lead to chronic stress that dysregulates the hypothalamic–pituitary–adrenal (HPA) axis, which—in excess—can lead to SPD [11,18]. Conversely, working in managerial roles and/or low-intensity industries with standard work hours (i.e., not overworked nor working long hours) and adequate compensation may allow for workplace autonomy, minimize work-related stress, and thus contribute to better mental health outcomes [11].

Given these potential pathways, race/ethnicity, gender, age, annual household income, and health status may influence the relationship between adverse occupational characteristics and SPD [11,19,20]. Due to institutional racism in the United States workforce, minoritized racial/ethnic groups, including Asian, NH-Black, and Hispanic/Latinx adults, are more likely to experience racial discrimination in hiring practices and while in the workplace [21,22]. For example, one study sent 83,000 fictitious job applications with randomized applicant characteristics (e.g., race) to 108 United States employers and concluded that over 20% of companies discriminate against Black applicants, and that this result varied by industry of employment [23]. Further, NH-Black and Hispanic/Latinxs are more likely to work in support service rather than managerial roles and labor-intensive rather than low-intensity industries and thus may be differentially impacted by occupational factors [24,25]. Similarly, institutionalized practices of sexism often translate to women being more likely to work in supportive roles, less likely to be considered for promotions, and more likely to be paid less than men in the same position despite the same level of educational attainment [17,26,27]. On average, women earn 18.5% less than men, and compared to NH-White men, NH-Black and Hispanic/Latinx women earn 32% and 38% less, respectively [24]. Although middle-aged and older adults are more likely than younger adults to hold management positions with more autonomy [17], they are more likely to have time-consuming and stressful responsibilities outside work (e.g., caring for aging parents while raising their own children), which may spill over into the workplace and negatively impact the relationship between occupational characteristics and SPD [28]. Furthermore, adults with lower compared to higher household incomes may be more likely to experience effort-reward imbalance (e.g., investing a lot of energy into a job with low pay) [29], which could contribute to a higher prevalence of SPD in adults with lower income. Also, adults with self-reported poor health, which may include people with disabilities and chronic illnesses, may be limited in employment opportunities due to unaccommodating and discriminatory work environments (i.e., ableism), and thus may be additionally burdened by occupational class and industry of employment [20] as well as SPD.

Despite the evidence for work-related disparities by race/ethnicity, gender, age, household income, and health status, few studies have examined the relationship between various industries of employment and occupational classes and SPD in a large sample of United States adults by these characteristics [8,9,10,11,12,16]. To address these important gaps in the literature, we investigated the association between industry of employment and occupational class and SPD overall and by race/ethnicity, gender, age, household income, and self-rated health status in a large, nationally representative sample. We hypothesized that the prevalence of SPD would vary by occupational class in that working in support services and laborer positions compared to professional/management positions would be associated with a higher prevalence of SPD in the overall population and within each industry of employment. We also hypothesized that the prevalence of SPD would vary by industry of employment, where participants working in industries (i.e., agriculture, retail, finance, education, healthcare, accommodations, public administration) other than the professional/administrative/management industry category would have a higher prevalence of SPD. Our research is grounded in the socioecological model, which posits that factors such as industry of employment and occupational class are influenced by upstream determinants such as structural racism, sexism, classism, and ableism [30]. Therefore, we also hypothesized that the association between occupational class and industry of employment and SPD would be stronger among minoritized (i.e., Asian, NH-Black, and Hispanic/Latinx participants) compared NH-White adults, women compared to men, middle aged and older compared to younger adults, adults with lower compared to higher household income, and adults with poor compared to good health.

## 2. Materials and Methods

### 2.1. Data Source: National Health Interview Survey

We analyzed data from the 2004–2018 National Health Interview Survey (NHIS), which is a series of cross-sectional, nationally representative surveys that uses a 3-stage stratified cluster probability sampling design to conduct interviews in households of non-institutionalized United States civilians. A detailed description of NHIS procedures is available elsewhere [31]. Briefly, trained interviewers from the United States Census Bureau used computer-assisted interviewing to conduct in-person interviews to obtain health information on each member of the sampled households. Each study participant provided informed consent, and the National Institute of Environmental Health Sciences Institutional Review Board waived approval for publicly available, secondary data with no identifiable information. The final response rate for sampled adults was 62.9% (range: 72.5% (2004)–53.1% (2018)).

### 2.2. Study Population

Our study population included adults aged ≥18 years who self-identified as Asian, NH-Black, Hispanic/Latinx, or NH-White. Participants were excluded from the analysis if they were unemployed/not in the labor force or had missing data on their employment status (n = 147,305), had missing data on their occupation (n = 38,910), had missing data on SPD (n = 8128), or if they self-identified as a race/ethnicity other than those listed above due to a small sample size (n = 5362). Our final analytic sample comprised 245,038 participants.

### 2.3. Exposure Assessment: Industry of Employment and Occupational Class

#### 2.3.1. Industry of Employment

Participants were asked if they were working the week prior to the survey and were included in the study if they worked at a paying or non-paying job or if they had a job or business but were not at work during the prior week. Afterwards, they were asked who they worked for (i.e., name of company, business, organization, or employer) and the kind of business or industry it was (e.g., retail shoe store, hospital). Participants were grouped into 21 major industry categories, and, as carried out in prior studies on employment industry [32,33], we utilized the North American Industry Classification System (NAICS) [34] codes to create the following eight broader industry groups: (1) professional, scientific, and technical services; management of companies and enterprises; and administrative and support and waste management and remediation services (referred to as professional/administrative/management); (2) agriculture, forestry, fishing, and hunting; mining; utilities; construction; manufacturing; wholesale trade; and transportation and warehousing (referred to as agriculture/manufacturing/construction); (3) retail trade; (4) information; finance and insurance; and real estate rental and leasing (referred to as finance/information/real estate); (5) educational services; (6) health care and social assistance; (7) accommodation and food services; and (8) other services (except public administration); public administration; and arts, entertainment, and recreation (referred to as public administration/arts/other services) [35,36] (Appendix A).

#### 2.3.2. Occupational Class

After determining if participants worked in the week prior to their interview or held a job, they were asked what kind of work they engaged in (e.g., car salesperson) and what were the most important activities on this job (e.g., sell cars). Participants were then grouped into 23 major occupational groups by the NHIS. We used the Standard Occupational Classification System [37] to separate these 23 groups in 3 occupational classes labeled: ‘Professional/Management’, ‘Support Services’, and ‘Laborers’ (Appendix A).

### 2.4. Outcome Assessment: Serious Psychological Distress

We used the Kessler Psychological Distress Scale (K6) to assess SPD. The K6 is a validated tool used to screen for serious mental illness that asks participants about six manifestations of nonspecific psychological distress [30]. Specifically, participants were asked the frequency with which they felt nervous, restless or fidgety, so sad that nothing could cheer you up, hopeless, that everything was an effort, and worthless in the 30 days leading up to their interview. Responses for each symptom were measured on a Likert scale from 0 (“None of the time”) to 4 (“All of the time”). Total scores ranged from 0 to 24 with higher scores representing higher levels of SPD. Consistent with the literature, participants with a total score ≥13 were categorized as having SPD [38].

### 2.5. Potential Confounders

Potential confounders, selected a priori, included examining self-reported sociodemographic, health behavior, and clinical factors. The following sociodemographic variables were considered confounders: age (18–30, 31–49, or ≥50 years); gender (woman or man), since it was unclear if participants responded based on biological sex or the social construct of gender; self-identified race/ethnicity (Asian, NH-Black, Hispanic/Latinx, or NH-White); marital status (married/living with partner/cohabitating, divorced/widowed/separated, or single/no live-in partner); educational attainment (<high school, high school graduate, or greater than high school); annual household income (<USD 35,000, USD 35,000–74,999, or ≥USD 75,000); and region of residence (Northeast, Midwest, South, or West). We also considered the following health behavior variables as confounders: smoking status (never/quit > 12 months prior, quit ≤ 12 months ago, or current), leisure-time physical activity (PA) based on recommended guidelines [39] (never/unable, does not meet PA guidelines, or meets PA guidelines), and alcohol consumption status (never, former, or current). Sleep health characteristics were not considered confounders because they are likely mediators between occupation and SPD [40]. Finally, we considered the following clinical characteristics as confounders: body mass index (BMI) (18.5–<25 kg/m^2^ (recommended), 25–29.9 kg/m^2^ (overweight), or ≥30 kg/m^2^ (obese)); self-rated health status (excellent/very good, good, or fair/poor); prior diagnosis (yes or no) of dyslipidemia, hypertension, and prediabetes/diabetes; and “ideal” cardiovascular health (yes or no), which included never/quit smoking >12 months prior to interview, meets leisure-time PA guidelines, BMI 18.5–<25 kg/m^2^, and has no prior diagnosis of dyslipidemia, hypertension, or diabetes/prediabetes [41].

### 2.6. Potential Modifiers: Race/Ethnicity, Gender, Age, Annual Household Income, and Self-Rated Health Status

We considered race/ethnicity (Asian, NH-Black, Hispanic/Latinx, NH-White), gender (woman or man), age (18–30, 31–49, ≥50 years), annual household income (<USD 75,000 or ≥USD 75,000), and self-rated health status (excellent/very good/good heath or fair/poor health) as potential modifiers of industry of employment and occupational class in relation to SPD.

### 2.7. Statistical Analysis

We calculated descriptive statistics of the study population overall, by occupational class, and by race/ethnicity. We expressed continuous variables as means ± standard errors (SE) and categorical variables as weighted percentages, which were age-standardized to the 2010 United States Census population. For all analyses, we used sampling weights, representing the inverse probability of selection into the sample, to account for unequal probability of selection resulting from the sampling design, nonresponse, and oversampling of older adults and minoritized racial/ethnic groups.

We used Poisson regression with robust variance to estimate prevalence ratios (PRs) and 95% confidence intervals (CIs) of SPD for each occupational class and employment industry [42]. We compared support services and laborer positions to professional/management positions overall and within each of the 8 industry categories. We also compared the 7 other industries of employment to the professional/administrative/management industry (henceforth referred to as the professional industry). We also stratified by race/ethnicity, gender, age, annual household income, and self-rated health status to test for differences in these associations. In supplemental analyses, we further stratified our gender, age, household income, and health status results by race/ethnicity. In each analysis, we adjusted for age, gender, race/ethnicity, marital/co-habiting status, educational attainment, annual household income, region of residence, self-rated health status, alcohol consumption status, and “ideal” cardiovascular health. Statistical significance was defined as a two-sided *p*-value of 0.05. We used STATA version 15 (STATA Corp., College Station, TX, USA) for all analyses.

## 3. Results

### 3.1. Study Population Characteristics

Of the 245,038 participants, 1.5% (n = 4606) were categorized as having SPD (Table 1). The mean age was 41.7 ± 0.1 years and 46.8% were women. The racial/ethnic composition of participants was 4.8% Asian, 10% NH-Black, 12.4% Hispanic/Latinx, and 72.8% NH-White. Further, 55.4% of participants had annual household incomes <USD 75,000 and 93.1% of participants reported excellent/very good/good health. Overall, 21.6% of participants held professional/management positions, 45.5% held support service positions, and 32.9% held laborer positions. Women were overrepresented among participants in support services positions (66.6%), and Hispanic/Latinx participants were overrepresented among laborers (20.6%). The portion of participants working in each industry of employment varied, with the highest percentage in agriculture/manufacturing/construction (27.2%) and the lowest in accommodations/food services (5.0%) (Appendix A).

The prevalence of SPD was highest among participants working in accommodation/food services (3.01%), retail trade (2.61%), and the health care/social assistance (2.36%) industries (Figure 1). Within the retail trade industry, Hispanic/Latinx (3.58%) and NH-Black (3.15%) participants had higher SPD than their NH-White and Asian counterparts. The prevalence of SPD was lowest among participants in professional/management positions (0.9%) and highest among laborers (1.7%) (Figure 2).

### 3.2. Occupational Class and Industry of Employment and Serious Psychological Distress Overall and by Race/Ethnicity

In the adjusted model, working in support services (PR = 1.19 [95% CI: 1.04–1.35]) and laborer (PR = 1.19 [95% CI: 1.04–1.38]) versus professional/management positions were both associated with 19% higher prevalence of SPD (Table 2). Working in other industries of employment versus the professional industry was associated with lower SPD: agriculture/manufacturing/construction (PR = 0.82 [95% CI: 0.70–0.95]); finance/information/real estate (PR = 0.72 [95% CI: 0.60–0.88]); educational services (PR = 0.79 [95% CI: 0.66–0.94]); and public administration/arts/other services (PR = 0.79 [95% CI: 0.67–0.92]). Within the health care/social assistance industry, working in a support service versus in a professional/management position was associated with 80% higher prevalence of SPD (PR = 1.80 [95% CI: 1.06–3.04]), after adjustment (Table 2).

Among Asian participants, working in the retail trade versus professional industry was associated with 2.68 times the prevalence of SPD (PR = 2.68 [95% CI: 1.24–5.77]) (Table 2). Among NH-Black participants in the education services industry, working in a support services position (versus in a professional/management position) was associated with nearly 9 times the prevalence of SPD (PR = 8.82 [95% CI: 1.09–71.2]) after adjustment. Among NH-White participants in the professional industry, working as a laborer versus in a professional/management position was associated with 53% higher SPD (PR = 1.53 [95% CI: 1.02–2.30]).

### 3.3. Occupational Class and Industry of Employment and Serious Psychological Distress by Gender

Among women, working as a laborer versus in a professional/management position was associated with 38% higher prevalence of SPD (PR = 1.38 [95% CI:1.14–1.67]), after adjustment (Table 3). The prevalence of SPD was lower in the agriculture/manufacturing/construction (PR = 0.81 [95% CI: 0.65–1.00]), finance/information/real estate (PR = 0.75 [95% CI: 0.59–0.96]), educational services (PR = 0.66 [95% CI: 0.53–0.84]), and public administration/arts/other services (PR = 0.78 [95% CI: 0.64–0.95]) industries compared to the professional industry among women and men.

Within the health care/social assistance industry, women who worked in support services and laborer versus professional/management positions had over twofold times the prevalence of SPD (PR = 2.29 [95% CI: 1.13–4.65]; PR = 2.41 [95% CI: 1.13–5.13]) (Table 3). Within the accommodation/food service industry, women who worked as laborers versus in professional/management positions also had twofold times the prevalence of SPD (PR = 2.10 [95% CI: 1.15–3.85]). Within the professional industry, men who worked in support services and laborer versus professional/management positions had a higher prevalence SPD (PR = 1.88 [95% CI: 1.20–2.94]; PR = 1.74 [95% CI: 1.10–2.73]). Gender–race/ethnicity-stratified associations for occupational class, industry of employment and SPD are shown in Appendix A. As one example, Asian men working in retail trade versus the professional industry had over 4 times the prevalence of SPD (PR = 4.10 [95% CI: 1.36–12.4]).

### 3.4. Occupational Class and Industry of Employment and Serious Psychological Distress by Age

Among participants 31–49 years old, working in a support service versus professional/management position was associated with 29% higher prevalence of SPD (PR = 1.29 [95% CI: 1.07–1.55]) and working as a laborer versus in a professional/management position was associated with 31% higher prevalence of SPD (PR = 1.31 [95% CI: 1.07–1.60]), after adjustment (Table 4). Among participants 18–30 years old and ≥50 years old, the prevalence of SPD was lower in the agriculture/manufacturing/construction (PR = 0.74 [95% CI: 0.57–0.96]), finance/information/real estate (PR = 0.62 [95% CI: 0.41–0.94]), health care/social assistance (PR = 0.62 [95% CI: 0.44–0.88]), and public administration/arts/other services (PR = 0.61 [95% CI: 0.45–0.84]) industries compared to the professional industry.

Among participants both 31–49 years old and ≥50 years old within the health care/social assistance industry, working in a support service versus professional/management position was associated with an over twofold times the prevalence of SPD, after adjustment. Among participants 31–49 years old in the accommodation/food services industry, working in a support services or laborer versus professional/management position was associated with over twofold times the prevalence of SPD, after adjustment (Table 4). Age-race/ethnicity-stratified associations for occupational class, industry of employment and SPD are dichotomized as ‘younger than 50 years’ or ‘at least 50 years or older’ (Appendix A).

### 3.5. Occupational Class and Industry of Employment and Serious Psychological Distress by Annual Household Income

Among participants with household incomes <USD 75,000, working in a support service versus professional/management position was associated with 19% higher prevalence of SPD (PR = 1.19 [95% CI: 1.01–1.4]) and working as a laborer versus in a professional/management position was associated with 22% higher prevalence of SPD (PR = 1.22 [95% CI: 1.03–1.45]), after adjustment (Table 5). In participants with household incomes <USD 75,000, the prevalence of SPD was lower in the agriculture/manufacturing/construction (PR = 0.79 [95% CI: 0.67–0.92]), finance/information/real estate (PR = 0.68 [95% CI: 0.55–0.83]), educational services (PR = 0.67 [95% CI: 0.55–0.82]), and public administration/arts/other services (PR = 0.77 [95% CI: 0.65–0.91]) industries compared to the professional industry.

Among participants working in the professional industry who had household incomes <USD 75,000, working in a support service versus professional/management position was associated with 46% higher prevalence of SPD (PR = 1.46 [95% CI: 1.04–2.05]) and working as a laborer versus in a professional/management position was associated with 59% higher prevalence of SPD (PR = 1.59 [95% CI: 1.11–2.29]), after adjustment (Table 5). Household income-race/ethnicity-stratified associations for occupational class, industry of employment and SPD are shown in Appendix A. Among NH-Black participants with household incomes <USD 75,000, working in support services and laborer versus professional/management positions overall and within the education services, health care/social assistance, and accommodation/food services industries was associated with very high SPD.

### 3.6. Occupational Class and Industry of Employment and Serious Psychological Distress by Self-Rated Health Status

Among participants reporting excellent, very good, and good health, working as a laborer versus in a professional/management position was associated with 23% higher prevalence of SPD (PR = 1.23 [95% CI: 1.04–1.46]), after adjustment (Table 6). Among participants in fair or poor health, working in a support services role versus in a professional/management position was associated with 34% higher prevalence of SPD (PR = 1.34 [95% CI: 1.01–1.78]), after adjustment. The prevalence of SPD was lower in the agriculture/manufacturing/construction (PR = 0.79 [95% CI: 0.67–0.95]), finance/information/real estate (PR = 0.64 [95% CI: 0.51–0.81]), educational services (PR = 0.67 [95% CI: 0.46–0.96]), and public administration/arts/other services (PR = 0.74 [95% CI: 0.62–0.89]) industries compared to the professional industry among participants with both excellent/very good/good health and fair/poor health status.

Among participants working in the health care/social assistance industry with fair/poor health, working in a support service versus professional/management position was associated with a near sixfold times the prevalence of SPD (PR = 5.98 [95% CI: 1.96–18.5]) and working as a laborer versus in a professional/management position was associated with a near fivefold times the prevalence of SPD (PR = 4.67 [95% CI: 1.34–16.3]), after adjustment (Table 6). Self-rated health status-race/ethnicity-stratified associations for occupational class, industry of employment and SPD are shown in Appendix A.

## 4. Discussion

In this large-scale nationally representative study, we investigated industry of employment and occupational class in relation to serious psychological distress. We also investigated potential modifiers of the relationship. Consistent with our hypothesis, we found that participants who worked in support services positions or as laborers had a higher prevalence of SPD when compared to participants who work in professional or management positions. However, individuals who worked in industries other than the professional industry category had lower SPD compared to participants in the professional industry, which was inconsistent with our hypothesis. When examining occupational class within each employment industry category, we found that working in a support services or laborer versus a professional/management position was associated with higher SPD in most industries. Regarding potential modifiers, these associations were stronger among: NH-Black compared to other racial/ethnic groups in the education services industry; women compared to men, middle aged and older compared to younger participants, lower compared to higher income participants in the professional industry; and participants in fair/poor health compared to those in at least good health in the health care/social assistance industry.

Our finding of high SPD in participants who worked in support services and laborer compared to professional/management positions is similar to prior literature. While few studies have examined this relationship in the US, a study of Dutch construction workers found that high work speed and quantity along with low participation in decision making (often correlated with support services or laborer positions) were associated with depressive symptoms [43]. Contrary to our hypothesis, we found that compared to participants who worked in the professional industry, individuals who worked in other industries had a lower prevalence of SPD; although estimates for the retail trade, health care/social assistance, and accommodation/food services industries were not statistically significant. There are a few potential explanations for these results. First, individuals in industries such as educational services and public administration/arts/other services may experience fulfillment or feel a sense of pride stemming from their job [44], which may help buffer stressful work situations. This was demonstrated in one study that found depression was significantly less prevalent among workers in amusement and recreation service industries (e.g., museum and garden operations) compared to the overall study population [45]. Further, it is possible that individuals who work in the professional industry are uniquely subjected to stress-inducing strain from, for instance, organizational bureaucracy, pressures from high decision-making latitude, combined with pressures to be highly productive [46]. More research is warranted. When examining occupational class within each employment industry, we found that within the health care/social assistance industry, participants in support service positions had higher SPD compared to those in professional/management positions. These findings support our hypothesis and may highlight stress-inducing factors that are unique to individuals in support service roles within these industries, such as the need to manage both people in lower and higher ranks within the organization, perceptions of powerlessness, and a potential lack of knowledge related to industry-specific ethics to make informed decisions [14].

While we did not observe effect modification by race/ethnicity in the associations between occupational class and SPD, we did observe strong associations within specific industries by race/ethnicity. For instance, Asian participants working in the retail trade compared to the professional industry had higher SPD while NH-Black participants in the education industry working in support services compared to professional/management positions had higher SPD. These findings may highlight systemic issues within many industries of employment, such as the exploitation (e.g., poor compensation, unpaid labor) of racially/ethnically minoritized individuals and discrimination in the workplace [21,22]. For example, minoritized racial/ethnic professors who are women have been shown to spend more time performing unpaid, emotional and service-oriented labor for students and their departments than White male professors [47,48]. This observation likely contributes to compromised mental health for structurally marginalized groups, which highlights a need to address adverse working conditions to alleviate public health burden. We also found that NH-White participants in the professional industry working as laborers had higher SPD compared to their counterparts in professional/management positions. This finding supports the notion that the exploitation of labor is also tied to class and classism and is not solely based upon race/ethnicity [49].

We observed several interesting findings related to the association between occupational class along with industry of employment and SPD by gender. First, we found that women who worked in support services and laborer positions within the health care/social assistance industry and in laborer positions in the accommodation/food services industry had higher SPD compared to women in professional/management positions in those industries. Supporting our hypothesis, these results expand the literature [15] by demonstrating an association by occupational class within specific employment industries. These results also may indicate that discriminatory and sexist practices, including workplace harassment, likely contribute to SPD in women. For example, employees at a Wendy’s in Weaverville, North Carolina, who went on strike due to sexual harassment from management, have been forced to find other work to sustain themselves, which can exacerbate SPD and contribute to gender-based health inequities [50].

There were also interesting findings among men. We found that men within the professional industry who worked in support services and laborer positions compared to professional/management positions had higher SPD. This finding may reflect the negative impact that patriarchy has on men, who, when faced with societal pressure to provide material and social resources for their families, may be demoralized from working in proximity to more highly paid and ranked employees [51]. This may be especially apparent within the professional industry, where there is a large pay gap between the highest and lowest paid employees [24]. We also found that Asian men in the retail trade compared to the professional industry had higher SPD. Although this estimate should be interpreted with caution due to the large confidence interval as a result of a small sample size, this finding demonstrates the importance of continuing to conduct occupational health and public health research in general at the intersections of social identities (e.g., race/ethnicity-gender).

Our age-stratified results indicated that the association between working in a support services or laborer position compared to a professional/management position and higher SPD was strongest among participants aged 31–49 years. We found a similar pattern by occupational class within two industries of employment: participants ages 31–49 years working in the accommodation/food services industry in support services or laborer positions, and participants ages 31–49 years and ≥50 years working in the health care/social assistance industry in support services positions had higher SPD compared to those in professional/management positions within those industries. These findings support our hypothesis and are similar to another study, which found that work stress is associated with elevated risk of prospective depressive symptoms in a sample of adults aged 50–64 years [52]. Further, these findings may highlight additional burdens faced by middle-aged adults, such as physically, emotionally, or financially caring for children and aging parents [28]. Future research studies should consider how occupation-related factors may be impacted by and contribute to stress in the home environment and implement/evaluate solutions that may improve public health (e.g., respite services), especially for middle-aged adults.

In contrast to our hypothesis, we did not find effect modification in the association between occupational class, industry of employment, and SPD by annual household income. Although our point estimates for participants with incomes <USD 75,000 were statistically significant, they were comparable to the estimates for participants with incomes ≥USD 75,000. It is possible that our smaller sample size among those with higher household incomes made it more difficult to make meaningful comparisons between these groups. However, we did find that among lower income participants in the professional industry, working in support services and laborer positions compared to professional/management positions was associated with higher SPD. As previously mentioned, this may highlight the potential mental health impact of wage gap issues within this industry including a large pay gap between the highest and lowest paid employees [24]. Increasing wages—at least to a livable wage—across industries can help reduce this public health burden and progress towards overall health equity. We also found that among NH-Black participants with lower household incomes, working in support services and laborer positions compared to professional/management positions was associated with higher SPD. This observation was apparent in the overall sample and in several industries of employment (e.g., education services). Although some of these estimates were not statistically significant, no studies, to our knowledge, have examined the relationship between occupational class in multiple industries of employment and SPD, so our results expand the literature in this area.

Participants in fair/poor health versus at least good health who worked in support services compared to professional/management positions had a higher prevalence of SPD, in alignment with our hypothesis and similar to previous literature [53]. Despite the significant overlap in the confidence intervals [54], this finding may have public health relevance by highlighting the need for more workplace interventions. Workplaces may more equitably accommodate job applicants and employees with chronic illnesses and disabilities by, for example, requiring disability awareness training and using adaptive technology in job applications [55]. However, when comparing participants in laborer positions to those in professional/management positions, we did not find differences by health status. There are a few explanations for these findings. First, very few participants reported being in fair/poor health, so it is possible that the small sample size in this group compared to those in at least good health made it difficult to detect meaningful differences. Further, a previous study that examined the relationship between poor sleep health and SPD found the association to be stronger among those in at least good health compared to those in fair/poor health [56]. The authors speculated that the relationship between sleep and SPD was less obscured among those in at least good health because there was an absence of people in poor health among this group. This may also be possible in our study. Similar to our other stratified analyses, the prevalence of SPD was lower among participants in several industries of employment compared to the professional industry in both health status groups. Among participants in fair/poor health in the health care/social assistance industry, participants working in support services and laborer positions compared to professional/management positions had higher SPD. This finding supports our hypothesis and is similar to a previous study that found physicians with versus without chronic illnesses are at a higher risk for depression [57].

Industry of employment and occupational class may impact SPD through myriad pathways [11]. Individuals who hold support service or labor-intensive positions may be burdened by long work hours, nonstandard work schedules, poor compensation, and harmful and oppressive working conditions [10,11,12]. Further, employees in these positions may not have decision-making latitude or the opportunity for upward social mobility, which can lead to demoralization and psychological stress [11,58]. Often, these positions do not provide benefits, such as paid-time off and other worker protections, forcing workers to lose pay in order to take care of themselves and their families, which likely adds additional stress [13]. This could lead to chronic stress, which may dysregulate the HPA axis [11]. Prolonged, unregulated, or excessive feelings of stress can lead to SPD [18]. Further, working in support and laborious positions in some industries (e.g., health care/social assistance) may be more stress-inducing than in other industries, so the mental health impact may differ across industries. Unlike what we hypothesized, we found that participants in professional industries had higher SPD than individuals in other industries (e.g., public administration). It is possible that employees in these industries deal with unique issues such as workplace bureaucracy and a lack of job fulfillment despite the expectation, which then may lead to SPD [44,45]. Additionally, these pathways may differ by sociodemographic groups, which is important to consider when creating equitable public health solutions. For example, middle-aged adults may be uniquely burdened by strain related to long work hours and/or receiving low compensation and being more likely to care for and financially support children as well as aging parents [28].

## 5. Limitations and Strengths

Our study has limitations. First, the cross-sectional design of the NHIS limits our ability to make causal inferences. Second, SPD was measured using self-reported data, and while the K6 scale is a validated scale for screening mental illness [38], it has only been validated across race/ethnicity in a sample of Californian adults [59]. Third, we were unable to estimate some associations between employment industry categories and SPD among Asian participants due to small sample sizes. Similarly, some estimates produced wide CIs that should be interpreted with caution. Fourth, while we used data from multiple study years (2004 to 2018), which reduces the risk of bias attributed to a single year of data collection, it is also possible that cultural shifts surrounding workplace culture and mental health could have impacted responses to questions over the study years. For example, the advent of new technologies (e.g., smartphones) may make it more difficult for people to separate their home and work life [60]. A focus on integration may be more necessary. Nevertheless, our large sample over 15 study years allowed us to examine and compare several industries of employment and occupational classes in a racially diverse sample that is representative of the United States populations included in the study. Further, we were able to stratify by multiple, important modifiers (i.e., race/ethnicity, gender, age, annual household income, and self-rated health status) to further investigate these associations within various sociodemographic groups given the large sample size. As such, this study adds to a growing body of literature regarding the role of specific employment industries and occupational classes on SPD among sociodemographic groups.

Future research studies should continue to assess the impact of occupational characteristics on mental health in these and other racial/ethnic groups (e.g., Native Americans), gender identity groups (e.g., nonbinary individuals), across sexual orientation groups, and across various types and levels of ability. Additionally, a focus on longitudinal and qualitative research is warranted to further unpack the stressors associated with working in various industries, occupational classes, and work environments, especially among populations with multiple minoritized identities in certain workplaces (e.g., Black women). Further, the interplay between the workplace and home environment as it relates to mental health should be further examined in groups that may be particularly burdened by their home environment (e.g., single parents, caregivers). Finally, research should continue to evaluate the implementation and effectiveness of various workplace mental health interventions, overall and within specific employment industries and occupational classes.

## 6. Conclusions

In this large nationally-representative study, working in support services and laborer positions compared to professional/management positions was associated with higher SPD. We also found that working in several industries of employment compared to the professional industry was associated with lower SPD. Although we did not find overall effect modification in our stratified analyses, the following examples depict several instances of high SPD among participants working in support services and laborer positions within specific industries: NH-Black participants in the education services industry; women in the health care/social assistance industry; middle-aged participants in the accommodation/food services industry; lower income participants in the professional industry; and participants in fair/poor health in the health care/social assistance industry. These findings may be important for informing future mental health-related interventions. Specifically, the work environment could serve as a promising intervention setting for reducing SPD. Implementing policies that make workplaces less taxing and more equitable may provide practical solutions to workplace-related mental illness. For example, increasing workplace flexibilities (where possible) such as maternal and parental leave, vacation time, and paid sick days may help relieve job-induced stress and make a broader range of positions and industries more accessible to a larger number of individuals in society [55,61]. Further, in addition to workplace interventions, policy changes on a national level (e.g., increasing federal minimum wage to a livable wage) will likely contribute to better mental health and reduce SPD across sociodemographic groups. Finally, shifting cultural norms in the US (e.g., remote work; implementing workplace flexibilities that enhance autonomy; incentives and accountability measures) may positively impact health in the overall population and in terms of equity.

## Figures and Tables

**Figure 1 ijerph-19-08376-f001:**
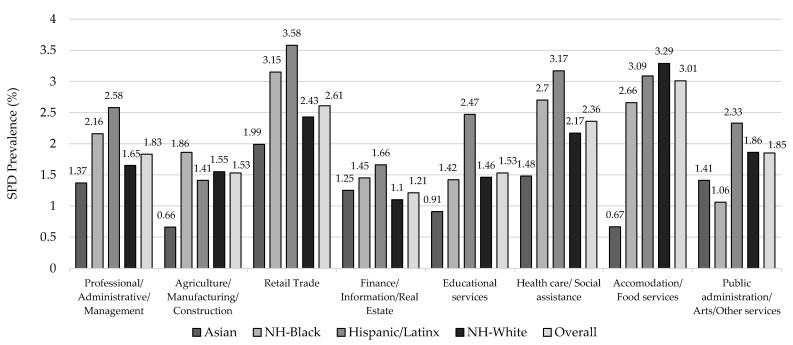
Age-Standardized ^a^ Prevalence of Serious Psychological Distress (SPD) by Industry of Employment among U.S. Adults Overall and by Race/Ethnicity, National Health Interview Survey, 2004–2018 (N = 245,038). ^a^ All estimates are weighted for the survey’s complex sampling design. All estimates except for age are age-standardized to the U.S. 2010 population. Professional/Administrative/Management includes the following NAICS (NAICS Association, LLC, Rockaway, NJ, USA) industry categories: professional, scientific, and technical services; management of companies and enterprises; and administrative support and waste management and remediation services industries. Agriculture/Manufacturing/Construction includes the following NAICS industry categories: agriculture, forestry, fishing, and hunting; mining; utilities; construction; manufacturing; wholesale trade; and transportation and warehousing. Finance/Information/Real Estate includes the following NAICS industry categories: information; finance and insurance; and real estate rental and leasing. Public administration/Arts/Other services include the following NAICS industry categories: public administration; arts, entertainment, and recreation; and other services (except public administration).

**Figure 2 ijerph-19-08376-f002:**
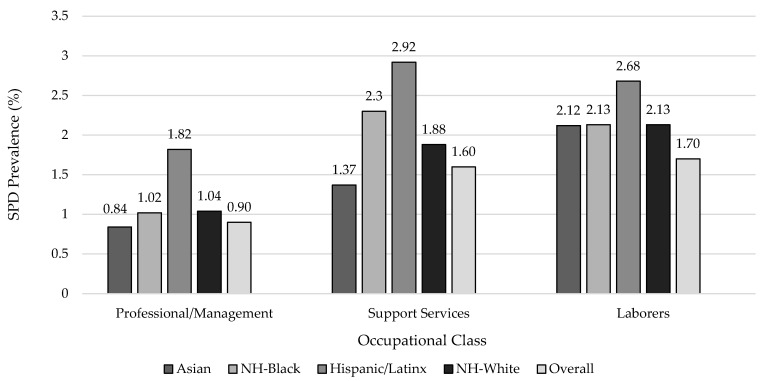
Age-Standardized ^a^ Prevalence of Serious Psychological Distress (SPD) by Occupational Class among U.S. Adults Overall and by Race/Ethnicity, National Health Interview Survey, 2004–2018 (N = 245,038). ^a^ All estimates are weighted for the survey’s complex sampling design. All estimates except for age are age-standardized to the U.S. 2010 population.

**Table 1 ijerph-19-08376-t001:** Age-Standardized ^a^ Sociodemographic, Health Behavior, and Clinical Characteristics among U.S. Adults Overall, by Occupational Class, and stratified by Serious Psychological Distress (SPD), National Health Interview Survey, 2004–2018, (N = 245,038).

Occupational Class										Total (N = 245,038)		
	Professional/Management n = 52,964 (21.6%)			Support Services n = 111,479 (45.5%)			Laborers n = 80,595 (32.9%)					
	All N = 52,964	No SPD N = 52,395	With SPD N = 569	All N = 111,479	No SPD N = 109,250	With SPD N = 2229	All N = 80,595	No SPD N = 78,787	With SPD N = 1808	All N = 245,038	No SPD N = 240,432	With SPD N = 4606
**SPD ^b^ (% yes)**	0.9			1.6			1.7			1.5		
**Sociodemographic Characteristics**												
**Age**, mean ± SE (years)	43.8 ± 0.1	43.8 ± 0.1	41.0 ± 0.7	41.5 ± 0.1	41.5 ± 0.1	39.6 ± 0.4	40.5 ± 0.1	40.5 ± 0.1	39.9 ± 0.4	41.7 ± 0.1	41.7 ± 0.1	39.9 ± 0.3
18–30	17.7	17.6	26.9	27.5	27.5	30.1	28.8	28.9	26.7	25.8	25.7	28.3
31–49	48.2	48.2	46.3	41.1	41.1	44.3	42.5	42.4	48.4	43.1	43.1	46.1
≥50	34.2	34.2	26.8	31.4	31.5	25.7	28.7	28.7	24.9	31.1	31.2	25.5
**Race/Ethnicity**												
Asian	6.9	6.9	5.4	4.8	4.8	3.3	3.5	3.5	3.3	4.8	4.8	3.5
NH-Black	6.3	6.3	6.0	10.6	10.6	12.2	11.9	11.9	11.3	10.0	10.0	11.0
Hispanic/Latinx	6.5	6.5	11.0	9.6	9.6	14.0	20.6	20.5	24.6	12.4	12.4	17.6
NH-White	80.3	80.3	77.6	75.0	75.1	70.6	64.0	64.0	60.8	72.8	72.8	67.9
**Women**	37.0	36.8	52.2	66.6	66.4	76.6	25.6	25.2	47.8	46.8	46.6	62.1
**Living in poverty ^c^**	1.7	1.6	4.0	5.2	5.1	12.9	9.0	8.8	21.9	5.6	5.5	14.7
**Other government assistance**	1.9	1.8	7.8	5.4	5.3	15.4	9.1	8.8	21.5	5.8	5.6	16.6
**Annual Household Income**												
<USD 35,000	8.4	8.2	22.9	21.6	21.3	42.9	32.5	32.1	53.6	22.0	21.7	44.0
USD 35,000–74,999	23.8	23.7	32.0	33.8	33.8	34.6	39.9	40.0	33.2	33.4	33.4	33.7
≥USD 75,000	67.8	68.0	45.0	44.5	44.9	22.5	27.7	27.9	13.2	44.6	45.0	22.3
**Educational Attainment**												
<High School	1.7	1.7	2.9	3.7	3.6	10.0	18.4	18.2	32.4	7.9	7.7	16.8
High School Graduate	12.7	12.7	12.5	22.9	22.8	26.8	41.8	41.7	41.2	26.5	26.4	30.2
Some College	23.9	23.9	32.1	33.5	33.4	39.4	29.7	29.8	22.1	30.1	30.1	31.9
≥College	61.6	61.7	52.4	39.9	40.2	23.8	10.1	10.2	4.3	35.5	35.7	21.1
**Industry of Employment**												
Professional/Administrative/Management ^d^	25.3	25.3	23.9	6.4	6.4	6.2	9.2	9.2	13.8	11.6	11.6	11.3
Agriculture/Manufacturing/Construction ^e^	28.1	28.1	27.4	11.5	11.6	9.8	49.1	49.3	35.1	27.2	27.3	22.2
Retail Trade	2.7	2.7	8.2	16.9	16.9	21.2	6.3	6.3	7.1	10.3	10.3	14.3
Finance/Information/Real Estate ^f^	16.1	16.2	15.2	10.2	10.3	6.8	2.9	2.9	1.8	9.3	9.3	6.0
Educational services	5.4	5.4	6.3	16.2	16.3	11	4.9	4.9	5.2	10.1	10.1	8.2
Health care/Social assistance	6.8	6.8	4.4	24.0	23.9	29.2	3.7	3.6	7.1	13.6	13.5	17.2
Accommodation/Food services	3.7	3.7	4.4	1.4	1.4	2.1	10.8	10.7	17.2	5.0	4.9	8.0
Public Administration/Arts/Other Services ^g^	11.9	11.9	10.2	13.3	13.3	13.7	13.1	13.2	12.8	13.0	13.0	12.8
**Marital/Co-habiting Status**												
Married/living with partner or cohabitating	72.6	72.8	55.8	63.4	63.7	45.2	64.9	65.1	54.1	65.9	66.1	49.7
Divorced/widowed/no live-in partner	15.7	15.6	27.8	20.5	20.3	34.9	19.1	18.9	28.3	19.0	18.8	31.7
Single/no live-in partner	11.7	11.7	16.4	16.1	16.1	19.8	16.0	16.0	17.7	15.1	15.1	18.6
**Region of residence**												
Northeast	19.6	19.6	17.1	19.7	19.8	14.3	17.2	17.2	14.7	18.9	18.9	14.8
Midwest	22.8	22.8	23.7	23.8	23.8	21.7	25.2	25.2	24.1	24.0	24.0	22.7
South	34.0	34.0	33.0	35.1	35.0	39.2	36.9	36.9	37.7	35.4	35.4	38.0
West	23.6	23.6	26.2	21.4	21.4	24.8	20.7	20.7	23.5	21.8	21.7	24.4
**Health Behaviors**												
**Smoking Status**												
Never/quit > 12 months prior *	88.1	88.3	75.1	85.2	85.4	72.1	75.5	75.8	60.8	82.8	83.0	68.3
Quit ≤ 12 months ago	1.1	1.1	2.0	1.2	1.2	1.7	1.7	1.7	2.1	1.3	1.3	1.9
Current	10.8	10.7	23.0	13.6	13.4	26.2	22.7	22.5	37.1	15.9	15.7	29.8
**Leisure-time physical activity**												
Never/Unable	21.7	21.6	32.9	28.8	28.6	43.4	42.6	42.4	51.4	31.5	31.3	44.7
Does not meet PA guidelines ^h^	19.7	19.7	28.2	21.1	21.1	21.3	17.2	17.2	18.7	19.5	19.5	21.6
Meets PA guidelines *	58.6	58.7	38.9	50.1	50.3	35.3	40.2	40.4	29.9	49.0	49.2	33.8
**Alcohol Status**												
Never	10.7	10.7	14.1	17.9	17.9	17	18.1	18.2	18	16.3	16.3	17.1
Former	10.1	10.1	15.1	12.4	12.3	19.6	16.8	16.8	20.6	13.2	13.1	19.1
Current	79.2	79.2	70.8	69.7	69.8	63.4	65.0	65.0	61.5	70.5	70.6	63.8
Heavy Alcohol Consumption	27.7	27.8	22.3	22.4	22.5	16.8	18.5	18.4	19.6	22.6	22.6	18.5
**Clinical Characteristics**												
Body Mass Index												
Recommended (18.5–<25 kg/m) *	33.4	33.4	30.1	36.5	36.5	32.2	27.6	27.6	29.2	32.9	32.9	30.6
Overweight (25–29.9 kg/m)	40.7	40.7	37.3	35.8	35.9	27.1	41.5	41.6	33.6	38.7	38.8	31.2
Obese (≥30 kg/m)	26.0	25.9	32.6	27.8	27.6	40.7	31.0	30.8	37.2	28.4	28.2	38.2
Health status												
Excellent/Very good/Good	95.5	95.7	74.3	93.8	94.2	63.5	90.2	90.6	63.1	93.1	93.5	67.2
Fair/Poor	4.5	4.3	25.7	6.2	5.8	31.7	9.8	9.4	36.9	6.9	6.5	32.8
Cancer (any)	11.0	10.9	19.0	9.6	9.6	12.0	7.2	7.1	12.0	9.2	9.1	13.1
Dyslipidemia (yes) ^i,^*	52.9	52.7	71.4	50.9	50.7	62.9	51.6	51.5	55.3	51.7	51.5	61.8
Hypertension (yes) *	30.4	30.3	43.7	31.2	31.0	43.7	33.7	33.4	48.3	31.7	31.5	45.3
Pre-Diabetes/Diabetes (yes) *	12.3	12.2	22.3	12.9	12.8	21.5	14.2	14.0	24.9	13.1	13.0	22.8
(Ideal) Cardiovascular Health (yes) ^j^	14.8	14.9	7.3	13.9	14.0	6.7	6.3	6.4	2.7	11.6	11.7	5.3

Abbreviations: kg = kilograms, m = meters, NH = non-Hispanic, PA = physical activity, SE = standard error, SPD = serious psychological distress. ^a^ All estimates are weighted for the survey’s complex sampling design. All estimates except for age are age-standardized to the U.S. 2010 population. Percentages may not sum to 100 due to missing values or rounding. ^b^ Kessler-6 psychological distress scale score ≥13. ^c^ <100% Federal Poverty Level. ^d^ Includes the following NAICS (NAICS Association, LLC, Rockaway, NJ, USA) industry categories: professional, scientific, and technical services; management of companies and enterprises; and administrative support and waste management and remediation services industries. ^e^ Includes the following NAICS industry categories: agriculture, forestry, fishing, and hunting; mining; utilities; construction; manufacturing; wholesale trade; and transportation and warehousing. ^f^ Includes the following NAICS industry categories: information; finance and insurance; and real estate rental and leasing. ^g^ Includes the following NAICS industry categories: public administration; arts, entertainment, and recreation; and other services (except public administration). ^h^ Meets PA guidelines defined as ≥150 min/week of moderate intensity or ≥75 min/week of vigorous intensity or ≥150 min/week of moderate + vigorous intensity physical activity. ^i^ Dyslipidemia defined as high cholesterol in the 12 months prior to interview. Available for survey years 2011–2018. ^j^ Ideal cardiovascular health includes never smoking/quit >12 months prior to interview, BMI 18.5–<25 kg/m, meeting physical activity guidelines, and no prior diagnosis of dyslipidemia, hypertension, or diabetes/prediabetes. * Indicator of ideal cardiovascular health.

**Table 2 ijerph-19-08376-t002:** Adjusted Prevalence Ratios (95% Confidence Intervals) for the Association Between Occupational Class (compared to Professional/Management positions) and Industry of Employment (compared to the Professional/Administrative/Management Industry) and Serious Psychological Distress, Stratified by Race/Ethnicity, National Health Interview Survey, 2004–2018, (N = 245,038).

	Overall (N = 245,038)	Race/Ethnicity			
		Asian (N = 13,488)	NH-Black (N = 32,164)	Hispanic/Latinx (N = 43,268)	NH-White (N = 156,118)
**Occupational Class ^a^**					
Support services	**1.19**	1.46	1.22	1.21	1.16
	**[1.04–1.35]**	[0.75–2.84]	[0.83–1.78]	[0.88–1.67]	[0.99–1.36]
Laborers	**1.19**	1.29	1.17	1.21	1.19
	**[1.04–1.38]**	[0.64–2.59]	[0.78–1.76]	[0.85–1.72]	[1.00–1.42]
**Industry of Employment ^b^**					
Agriculture/Manufacturing/Construction	**0.82**	1.10	0.73	0.78	0.83
	**[0.70–0.95]**	[0.49–2.47]	[0.47–1.14]	[0.56–1.08]	[0.69–1.00]
Retail Trade	0.96	**2.68**	0.89	0.91	0.90
	[0.82–1.12]	**[1.24–5.77]**	[0.54–1.45]	[0.62–1.34]	[0.74–1.11]
Finance/Information/Real Estate	**0.72**	1.03	**0.52**	0.93	**0.69**
	**[0.60–0.88]**	[0.45–2.35]	**[0.30–0.92]**	[0.57–1.51]	**[0.55–0.88]**
Educational Services	**0.79**	1.52	0.67	0.77	**0.78**
	**[0.66–0.94]**	[0.58–3.97]	[0.39–1.16]	[0.47–1.27]	**[0.62–0.98]**
Health care/Social Assistance	0.91	1.79	0.72	1.05	0.87
	[0.78–1.05]	[0.76–4.24]	[0.46–1.13]	[0.75–1.47]	[0.72–1.05]
Accommodation/Food services	0.97	**0.37**	0.79	0.96	1.02
	[0.82–1.15]	**[0.15–0.93]**	[0.46–1.35]	[0.68–1.36]	[0.82–1.27]
Public Administration/Arts/Other services	**0.79**	1.43	**0.55**	0.77	**0.81**
	**[0.67–0.92]**	[0.68–3.03]	**[0.34–0.88]**	[0.52–1.15]	**[0.67–0.98]**
Professional/Administrative/Management					
Support services	1.32	NE	1.32	1.07	1.47
	[0.96–1.82]		[0.45–3.85]	[0.52–2.20]	[1.00–2.16]
Laborers	1.40	NE	1.23	1.09	**1.53**
	[0.99–1.99]		[0.43–3.54]	[0.46–2.56]	**[1.02–2.30]**
Manufacturing/Construction					
Support services	1.06	**0.20**	0.98	1.07	1.14
	[0.76–1.46]	**[0.04–0.97]**	[0.39–2.44]	[0.51–2.23]	[0.75–1.72]
Laborers	1.12	0.65	1.00	0.89	1.21
	[0.85–1.48]	[0.23–1.89]	[0.38–2.60]	[0.48–1.63]	[0.86–1.70]
Retail Trade					
Support services	1.29	NE	1.22	0.70	1.30
	[0.73–2.28]		[0.22–6.65]	[0.19–2.54]	[0.69–2.46]
Laborers	1.44	NE	1.02	1.25	1.28
	[0.77–2.70]		[0.16–6.42]	[0.32–4.82]	[0.61–2.68]
Finance/Information/Real Estate					
Support services	0.93	NE	NE	1.83	0.86
	[0.64–1.34]			[0.75–4.46]	[0.55–1.34]
Laborers	0.83	NE	NE	1.71	0.79
	[0.49–1.41]			[0.58–5.04]	[0.39–1.60]
Educational Services					
Support services	1.24	NE	**8.82**	0.60	1.13
	[0.73–2.11]		**[1.09–71.2]**	[0.15–2.45]	[0.62–2.04]
Laborers	0.97	NE	7.65	**0.19**	0.99
	[0.53–1.76]		[0.77–75.5]	**[0.04–0.99]**	[0.49–1.99]
Healthcare/Social assistance					
Support services	**1.80**	0.86	2.39	2.80	1.54
	**[1.06–3.04]**	[0.10–7.07]	[0.58–9.91]	[0.92–8.57]	[0.81–2.94]
Laborers	1.63	1.10	2.32	1.53	1.39
	[0.90–2.94]	[0.06–21.7]	[0.52–10.3]	[0.40–5.80]	[0.65–3.01]
Accommodation/Food Services					
Support services	1.37	NE	NE	0.75	1.45
	[0.81–2.32]			[0.25–2.26]	[0.77–2.72]
Laborers	1.49	NE	NE	1.25	1.38
	[0.94–2.38]			[0.54–2.89]	[0.78–2.44]
Public Administration/Arts/Other Services					
Support services	1.25	0.89	1.36	1.11	1.26
	[0.87–1.80]	[0.30–2.65]	[0.57–3.26]	[0.32–3.84]	[0.81–1.96]
Laborers	1.11	2.16	0.87	1.68	0.95
	[0.73–1.69]	[0.50–9.37]	[0.31–2.44]	[0.45–6.26]	[0.56–1.61]

Abbreviations: NH = non-Hispanic NE = not estimable. ^a^ Reference group: professional/management positions. ^b^ Reference group: professional/administrative/management industry. All models adjusted for age (18–30, 31–49, ≥50 years), gender (women, men), educational attainment (<high school, high school graduate, some college, ≥college), annual household income (<USD 35,000, USD 35,000–74,999, USD 75,000+), region of residence (Northeast, Midwest, South, West), marital/co-habiting status (married/living with partner or cohabitating, divorced/widowed/separated, single/no live-in partner), health status (excellent/very good, good, fair/poor), alcohol consumption (never, former, current), and “ideal” cardiovascular health (never smoking/quit >12 months prior to interview, BMI 18.5–<25 kg/m, meeting physical activity guidelines, and no prior diagnosis of dyslipidemia, hypertension, or diabetes/prediabetes). Overall models also adjusted for race/ethnicity (NH-White, NH-Black, Hispanic/Latinx, and Asian). Boldface indicates statistically significant results at the 0.05 level.

**Table 3 ijerph-19-08376-t003:** Adjusted Prevalence Ratios (95% Confidence Intervals) for the Association Between Occupational Class (compared to Professional/Management positions) and Industry of Employment (compared to the Professional/Administrative/Management Industry) and Serious Psychological Distress, Stratified by Gender, National Health Interview Survey, 2004–2018 (N = 245,038).

Gender	Women (N = 123,634)	Men (N = 121,404)
**Occupational Class ^a^**		
Support services	**1.22**	1.23
	**[1.03–1.45]**	[0.99–1.53]
Laborers	**1.38**	1.05
	**[1.14–1.67]**	[0.86–1.30]
**Industry of Employment ^b^**		
Agriculture/Manufacturing/Construction	0.85	**0.81**
	[0.69–1.05]	**[0.65–1.00]**
Retail Trade	0.96	0.94
	[0.78–1.18]	[0.73–1.22]
Finance/Information/Real Estate	**0.75**	**0.67**
	**[0.59–0.96]**	**[0.48–0.94]**
Educational Services	**0.66**	1.21
	**[0.53–0.84]**	[0.88–1.67]
Healthcare/Social Assistance	0.87	1.21
	[0.72–1.05]	[0.88–1.65]
Accommodation/Food services	1.07	0.76
	[0.87–1.32]	[0.56–1.02]
Public Administration/Arts/Other services	**0.78**	0.78
	**[0.64–0.95]**	[0.60–1.01]
Professional/Administrative/Management		
Support services	1.08	**1.88**
	[0.70–1.67]	**[1.20–2.94]**
Laborers	1.16	**1.74**
	[0.72–1.87]	**[1.10–2.73]**
Agriculture/Manufacturing/Construction		
Support services	1.13	1.04
	[0.74–1.73]	[0.62–1.74]
Laborers	1.24	1.04
	[0.82–1.85]	[0.72–1.52]
Retail Trade		
Support services	1.53	1.10
	[0.70–3.32]	[0.45–2.67]
Laborers	1.66	1.21
	[0.68–4.07]	[0.47–3.11]
Finance/Information/Real Estate		
Support services	1.01	0.67
	[0.64–1.59]	[0.35–1.29]
Laborers	0.48	0.95
	[0.20–1.14]	[0.46–1.94]
Educational Services		
Support services	1.03	1.66
	[0.52–2.05]	[0.74–3.73]
Laborers	1.23	0.61
	[0.54–2.82]	[0.24–1.57]
Health Care/Social Assistance		
Support services	**2.29**	1.03
	**[1.13–4.65]**	[0.46–2.33]
Laborers	**2.41**	0.57
	**[1.13–5.13]**	[0.22–1.53]
Accommodation/Food Services		
Support services	1.79	0.89
	[0.94–3.43]	[0.34–2.34]
Laborers	**2.10**	0.63
	**[1.15–3.85]**	[0.29–1.37]
Public Administration/Arts/Other Services		
Support services	1.34	1.08
	[0.85–2.10]	[0.58–2.02]
Laborers	1.32	0.91
	[0.77–2.28]	[0.47–1.76]

^a^ Reference group: professional/management positions. ^b^ Reference group: professional/administrative/management industry. All models adjusted for age (18–30, 31–49, ≥50 years), gender (women, men), educational attainment (<high school, high school graduate, some college, ≥college), annual household income (<USD 35,000, USD 35,000–74,999, USD 75,000+), region of residence (Northeast, Midwest, South, West), marital/co-habiting status (married/living with partner or cohabitating, divorced/widowed/separated, single/no live-in partner), health status (excellent/very good, good, fair/poor), alcohol consumption (never, former, current), and “ideal” cardiovascular health (never smoking/quit >12 months prior to interview, BMI 18.5–<25 kg/m, meeting physical activity guidelines, and no prior diagnosis of dyslipidemia, hypertension, or diabetes/prediabetes). All estimates are weighted for the survey’s complex sampling design. Boldface indicates statistically significant results at the 0.05 level.

**Table 4 ijerph-19-08376-t004:** Adjusted Prevalence Ratios (95% Confidence Intervals) for the Association Between Occupational Class (compared to Professional/Management positions) and Industry of Employment (compared to the Professional/Administrative/Management Industry) and Serious Psychological Distress, Stratified by Age, National Health Interview Survey, 2004–2018, (N = 245,038).

Age	18–30 Years (N = 57,780)	31–49 Years (N = 106,431)	≥50 Years (N = 80,827)
**Occupational Class ^a^**			
Support services	0.95	**1.29**	1.13
	[0.71–1.28]	**[1.07–1.55]**	[0.88–1.44]
Laborers	0.86	**1.3**	1.21
	[0.62–1.19]	**[1.07–1.60]**	[0.95–1.55]
**Industry of Employment ^b^**			
Agriculture/Manufacturing/Construction	0.76	0.89	**0.74**
	[0.55–1.06]	[0.72–1.10]	**[0.57–0.96]**
Retail Trade	0.94	0.97	0.84
	[0.71–1.26]	[0.77–1.23]	[0.61–1.15]
Finance/Information/Real Estate	**0.62**	0.80	0.71
	**[0.41–0.94]**	[0.61–1.05]	[0.49–1.02]
Educational Services	0.74	0.84	0.76
	[0.49–1.10]	[0.63–1.11]	[0.55–1.07]
Healthcare/Social Assistance	**0.62**	1.01	0.99
	**[0.44–0.88]**	[0.82–1.25]	[0.74–1.31]
Accommodation/Food services	0.92	1.08	0.67
	[0.68–1.23]	[0.85–1.39]	[0.45–1.01]
Public Administration/Arts/Other services	0.74	0.92	**0.61**
	[0.53–1.03]	[0.74–1.16]	**[0.45–0.84]**
Professional/Administrative/Management			
Support services	1.31	**1.54**	0.88
	[0.62–2.75]	**[1.03–2.32]**	[0.47–1.61]
Laborers	1.11	1.44	1.37
	[0.49–2.50]	[0.88–2.36]	[0.77–2.42]
Agriculture/Manufacturing/Construction			
Support services	1.07	1.03	0.98
	[0.41–2.77]	[0.67–1.59]	[0.61–1.57]
Laborers	0.89	1.20	1.02
	[0.37–2.16]	[0.84–1.71]	[0.68–1.54]
Retail Trade			
Support services	1.23	2.32	0.51
	[0.47–3.20]	[0.91–5.90]	[0.21–1.25]
Laborers	1.21	2.36	0.68
	[0.43–3.42]	[0.84–6.62]	[0.23–1.97]
Finance/Information/Real Estate			
Support services	0.94	1.12	0.62
	[0.43–2.02]	[0.67–1.87]	[0.32–1.20]
Laborers	0.20	0.97	0.78
	[0.04–1.12]	[0.42–2.23]	[0.34–1.81]
Educational Services			
Support services	0.53	1.78	2.36
	[0.22–1.26]	[0.75–4.25]	[0.82–6.8]
Laborers	0.40	1.02	2.64
	[0.11–1.47]	[0.37–2.77]	[0.83–8.36]
Health Care/Social Assistance			
Support services	0.75	**2.23**	**2.26**
	[0.21–2.69]	**[1.16–4.27]**	**[1.08–4.75]**
Laborers	0.35	**2.16**	2.39
	[0.07–1.71]	**[1.00–4.69]**	[0.98–5.83]
Accommodation/Food Services			
Support services	0.73	**2.51**	2.64
	[0.34–1.57]	**[1.22–5.18]**	[0.58–12.0]
Laborers	0.72	**2.69**	3.49
	[0.38–1.39]	**[1.40–5.19]**	[0.86–14.1]
Public Administration/Arts/Other Services			
Support services	1.93	1.10	0.97
	[0.85–4.35]	[0.65–1.87]	[0.48–1.95]
Laborers	0.98	1.12	1.08
	[0.36–2.71]	[0.61–2.08]	[0.52–2.25]

^a^ Reference group: professional/management positions. ^b^ Reference group: professional/administrative/management industry. All models adjusted for age (18–30, 31–49, ≥50 years), gender (women, men), educational attainment (<high school, high school graduate, some college, ≥college), annual household income (<USD 35,000, USD 35,000–74,999, USD 75,000+), region of residence (Northeast, Midwest, South, West), marital/co-habiting status (married/living with partner or cohabitating, divorced/widowed/separated, single/no live-in partner), health status (excellent/very good, good, fair/poor), alcohol consumption (never, former, current), and “ideal” cardiovascular health (never smoking/quit >12 months prior to interview, BMI 18.5–<25 kg/m, meeting physical activity guidelines, and no prior diagnosis of dyslipidemia, hypertension, or diabetes/prediabetes). All estimates are weighted for the survey’s complex sampling design. Boldface indicates statistically significant results at the 0.05 level.

**Table 5 ijerph-19-08376-t005:** Adjusted Prevalence Ratios (95% Confidence Intervals) for the Association Between Occupational Class (compared to Professional/Management positions) and Industry of Employment (compared to the Professional/Administrative/Management Industry) and Serious Psychological Distress, Stratified by Annual Household Income, National Health Interview Survey, 2004–2018, (N = 226,858) ^a^.

Annual Household Income	<USD 75,000 (N = 146,865)	≥USD 75,000 (N = 79,993)
**Occupational Class ^b^**		
Support services	**1.19**	1.19
	**[1.01–1.4]**	[0.94–1.51]
Laborers	**1.22**	1.14
	**[1.03–1.45]**	[0.84–1.56]
**Industry of Employment ^c^**		
Agriculture/Manufacturing/Construction	**0.79**	0.80
	**[0.67–0.92]**	[0.56–1.14]
Retail Trade	0.93	0.93
	[0.79–1.11]	[0.61–1.41]
Finance/Information/Real Estate	**0.68**	0.76
	**[0.55–0.83]**	[0.49–1.18]
Educational Services	**0.67**	1.09
	**[0.55–0.82]**	[0.73–1.62]
Healthcare/Social Assistance	0.91	0.84
	[0.78–1.07]	[0.57–1.23]
Accommodation/Food services	0.99	0.93
	[0.84–1.17]	[0.55–1.59]
Public Administration/Arts/Other services	**0.77**	0.78
	**[0.65–0.91]**	[0.54–1.14]
Professional/Administrative/Management		
Support services	**1.46**	1.06
	**[1.04–2.05]**	[0.56–2.00]
Laborers	**1.59**	1.05
	**[1.11–2.29]**	[0.44–2.50]
Agriculture/Manufacturing/Construction		
Support services	0.97	1.23
	[0.66–1.43]	[0.69–2.20]
Laborers	1.16	1.03
	[0.80–1.68]	[0.66–1.61]
Retail Trade		
Support services	1.31	1.15
	[0.67–2.56]	[0.40–3.27]
Laborers	1.33	1.55
	[0.64–2.76]	[0.46–5.26]
Finance/Information/Real Estate		
Support services	0.87	1.00
	[0.57–1.32]	[0.53–1.91]
Laborers	0.77	1.09
	[0.42–1.39]	[0.35–3.37]
Educational Services		
Support services	0.91	2.00
	[0.49–1.68]	[0.74–5.46]
Laborers	0.99	0.68
	[0.51–1.94]	[0.16–2.90]
Health Care/Social Assistance		
Support services	1.58	2.37
	[0.85–2.95]	[0.95–5.89]
Laborers	1.39	3.53
	[0.70–2.75]	[1.03–12.1]
Accommodation/Food Services		
Support services	1.36	1.49
	[0.78–2.36]	[0.25–8.83]
Laborers	1.34	2.79
	[0.83–2.17]	[0.58–13.4]
Public Administration/Arts/Other Services		
Support services	1.41	1.06
	[0.88–2.25]	[0.57–1.97]
Laborers	1.19	1.24
	[0.71–1.98]	[0.60–2.55]

^a^ 18,180 participants excluded for missing income data. ^b^ Reference group: professional/management positions. ^c^ Reference group: professional/administrative/management industry. All models adjusted for age (18–30, 31–49, ≥50 years), gender (women, men), educational attainment (<high school, high school graduate, some college, ≥college), annual household income (<USD 35,000, USD 35,000–74,999, USD 75,000+), region of residence (Northeast, Midwest, South, West), marital/co-habiting status (married/living with partner or cohabitating, divorced/widowed/separated, single/no live-in partner), health status (excellent/very good, good, fair/poor), alcohol consumption (never, former, current), and “ideal” cardiovascular health (never smoking/quit >12 months prior to interview, BMI 18.5–<25 kg/m, meeting physical activity guidelines, and no prior diagnosis of dyslipidemia, hypertension, or diabetes/prediabetes). All estimates are weighted for the survey’s complex sampling design. Boldface indicates statistically significant results at the 0.05 level.

**Table 6 ijerph-19-08376-t006:** Adjusted Prevalence Ratios (95% Confidence Intervals) for the Association Between Occupational Class (compared to Professional/Management positions) and Industry of Employment (compared to the Professional/Administrative/Management Industry) and Serious Psychological Distress, Stratified by Self-Reported Health Status, National Health Interview Survey, 2004–2018, (N = 245,038).

Self-Reported Health Status	Excellent/Very Good/Good (N = 229,364)	Fair/Poor (N = 15,620)
**Occupational Class ^a^**		
Support services	1.15	**1.34**
	[0.99–1.34]	**[1.01–1.78]**
Laborers	**1.23**	1.21
	**[1.04–1.46]**	[0.91–1.61]
**Industry of Employment ^b^**		
Agriculture/Manufacturing/Construction	**0.79**	0.88
	**[0.67–0.95]**	[0.66–1.16]
Retail Trade	0.96	0.98
	[0.79–1.16]	[0.72–1.32]
Finance/Information/Real Estate	**0.64**	0.99
	**[0.51–0.81]**	[0.69–1.44]
Educational Services	0.81	**0.67**
	[0.66–1.00]	**[0.46–0.96]**
Healthcare/Social Assistance	0.85	1.10
	[0.71–1.01]	[0.82–1.47]
Accommodation/Food services	0.96	1.02
	[0.79–1.18]	[0.74–1.39]
Public Administration/Arts/Other services	**0.74**	0.89
	**[0.62–0.89]**	[0.67–1.19]
Professional/Administrative/Management		
Support services	1.30	1.55
	[0.89–1.90]	[0.82–2.94]
Laborers	1.42	1.68
	[0.93–2.15]	[0.88–3.19]
Agriculture/Manufacturing/Construction		
Support services	0.95	1.54
	[0.65–1.39]	[0.81–2.92]
Laborers	1.10	1.32
	[0.79–1.52]	[0.77–2.27]
Retail Trade		
Support services	1.74	0.61
	[0.88–3.43]	[0.21–1.78]
Laborers	1.92	0.67
	[0.93–3.95]	[0.20–2.29]
Finance/Information/Real Estate		
Support services	0.97	0.74
	[0.64–1.49]	[0.40–1.38]
Laborers	0.81	0.75
	[0.39–1.68]	[0.32–1.77]
Educational Services		
Professional/Management	Ref	Ref
Support services	1.39	0.97
	[0.75–2.59]	[0.37–2.58]
Laborers	1.12	0.90
	[0.56–2.24]	[0.29–2.76]
Health Care/Social Assistance		
Support services	1.38	**5.98**
	[0.79–2.4]	**[1.93–18.5]**
Laborers	1.46	**4.67**
	[0.76–2.79]	**[1.34–16.3]**
Accommodation/Food Services		
Support services	1.25	2.63
	[0.69–2.25]	[0.76–9.13]
Laborers	1.35	2.84
	[0.82–2.24]	[0.91–8.92]
Public Administration/Arts/Other Services		
Support services	1.25	1.09
	[0.84–1.86]	[0.49–2.43]
Laborers	1.36	0.63
	[0.87–2.15]	[0.26–1.50]

^a^ Reference group: professional/management positions. ^b^ Reference group: professional/administrative/management industry. All models adjusted for age (18–30, 31–49, ≥50 years), gender (women, men), educational attainment (<high school, high school graduate, some college, ≥college), annual household income (<USD 35,000, USD 35,000–74,999, USD 75,000+), region of residence (Northeast, Midwest, South, West), marital/co-habiting status (married/living with partner or cohabitating, divorced/widowed/separated, single/no live-in partner), health status (excellent/very good, good, fair/poor), alcohol consumption (never, former, current), and “ideal” cardiovascular health (never smoking/quit >12 months prior to interview, BMI 18.5–<25 kg/m, meeting physical activity guidelines, and no prior diagnosis of dyslipidemia, hypertension, or diabetes/prediabetes). All estimates are weighted for the survey’s complex sampling design. Boldface indicates statistically significant results at the 0.05 level.

## Data Availability

The datasets generated during and/or analyzed during the current study are publicly available on the Integrated Health Interview Series Website (https://nhis.ipums.org/nhis/ (accessed on 1 December 2020)).

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
