# Peer review of "Employment Industry and Occupational Class in Relation to Serious Psychological Distress in the United States"

_ijerph, 2022, doi:10.3390/ijerph19148376_

Round 1

Reviewer 1 Report

Simply: perfect study!

Congratulations to authors.

Author Response

Reviewer #1: Simply: perfect study!

Congratulations to authors.

RESPONSE: Thank you for taking the time to review our study.

Reviewer 2 Report

Dear authors,

Thank you for allowing me to read the manuscript addressing whether occupational characteristics may influence serious psychological distress. This is particularly the case with the United States, which has been analyzed in your study.

As far as I understood from the paper, your research aims to provide an interesting approach to investigating the employment industry and occupational class in relation to SPD among racially/ethnically diverse adults in the US and by race/ethnicity, sex/gender, age, household income, and health status.

It appears that the research has been well designed and the paper is properly drafted in English with good academic soundness. However, a few aspects should be improved a little bit to better understand the whole manuscript. I hope your efforts in the revision of the manuscript according to my comments can make the paper more attractive to satisfy the high demands set by readers of the International Journal of Environmental Research and Public Health  (ISSN 1660-4601),

#Abstract:

Lines 31-32: Be aware that all words reflected in keywords must be previously quoted from the abstract in order of appearance. In that way, the terms Occupations and Economic status have not been reflected in the abstract. Please correct this as soon as possible.

#Introduction:

No objections.

#Method:

This paragraph is a key part of the manuscript. I also think that Table 1 is very valuable since they help to gain a deeper understanding of particular issues from your research.

#Results:

This subpart is good. Particularly commendable is the creation of Tables 2 and 3. That´s why I do not have anything negative to say about it. Moreover, Figures 1 and 2 are very helpful for readers, since they have summarized the most important numerical findings of the research.

#Discussion and conclusions:

This section seems too short for the ambitious objectives pointed out in the introduction. Please try to expand it to better understand the findings from this study and how they may provide a basis for further work. I truly believe that you should devote some paragraphs to consideration of matters relating to conclusions in a specific subchapter that would be separated from the discussion.

Additionally, I suggest that a separate section should be devoted to explaining your research recommendations for future studies.

Finally, I really cannot see any relationship between the findings of your study and the aims thereof in terms of public health. In fact, in all subchapters of the manuscript, the term public health is not once mentioned. Therefore, I encourage you to review the content of your findings to favor an approach toward public health.

Best Regards,

The Reviewer

Author Response

Reviewer #2: Thank you for allowing me to read the manuscript addressing whether occupational characteristics may influence serious psychological distress. This is particularly the case with the United States, which has been analyzed in your study.

As far as I understood from the paper, your research aims to provide an interesting approach to investigating the employment industry and occupational class in relation to SPD among racially/ethnically diverse adults in the US and by race/ethnicity, sex/gender, age, household income, and health status.

It appears that the research has been well designed and the paper is properly drafted in English with good academic soundness. However, a few aspects should be improved a little bit to better understand the whole manuscript. I hope your efforts in the revision of the manuscript according to my comments can make the paper more attractive to satisfy the high demands set by readers of the International Journal of Environmental Research and Public Health (ISSN 1660-4601).

RESPONSE: Thank you for your comments regarding our manuscript.

  1. #Abstract: Lines 31-32: Be aware that all words reflected in keywords must be previously quoted from the abstract in order of appearance. In that way, the terms “Occupations” and “Economic status” have not been reflected in the abstract. Please correct this as soon as possible.

RESPONSE: Thank you for this suggestion. We have reordered the key terms to the order that they appear in the abstract, and it does not appear to be a journal requirement for all keywords to be exactly quoted from the abstract, just highly relevant to the article in general. The text is provided below for your reference:

Line 33-34: “Keywords: Occupations; Occupational health; Psychological distress; Employment; Race factors; Economic status; Mental health”

  1. #Introduction: No objections.

RESPONSE: Thank you for your comment.

  1. #Method: This paragraph is a key part of the manuscript. I also think that Table 1 is very valuable since they help to gain a deeper understanding of particular issues from your research.

RESPONSE: Thank you for your comment.

  1. #Results: This subpart is good. Particularly commendable is the creation of Tables 2 and 3. That´s why I do not have anything negative to say about it. Moreover, Figures 1 and 2 are very helpful for readers, since they have summarized the most important numerical findings of the research.

RESPONSE: Thank you for your comment.

  1. #Discussion and conclusions: This section seems too short for the ambitious objectives pointed out in the introduction. Please try to expand it to better understand the findings from this study and how they may provide a basis for further work.

RESPONSE: We have added additional language throughout several paragraphs in the discussion to further interpret the findings in the context of public health and future research. The additional text is provided below for your reference:

Line 641-644: “This observation likely contributes to compromised mental health for structurally-marginalized groups, which highlights a need to address adverse working conditions to alleviate public health burden.”

Line 668-672: “Although this estimate should be interpreted with caution due to the large confidence interval as a result of a small sample size, this finding demonstrates the importance of continuing to conduct occupational health and public health research in general at the intersections of social identities (e.g., race/ethnicity-gender).”

Line 704-707: “Future research studies should consider how occupation-related factors may be impacted by and contribute to stress in the home environment and implement/evaluate solutions that may improve public health (e.g., respite services), especially for middle-aged adults.”

Line 718-720: “Increasing wages - at least to a livable wage - across industries can help reduce this public health burden and progress towards overall health equity.”

Line 859-862: “Finally, shifting cultural norms in the United States (e.g., implementing workplace flexibilities that enhance autonomy; incentives and accountability measures) may positively impact health in the overall population and in terms of equity.”  

  1. I truly believe that you should devote some paragraphs to consideration of matters relating to conclusions in a specific subchapter that would be separated from the discussion.

RESPONSE: As requested, we have added additional language in the conclusion section to further detail our findings in a broader public health context. The text is provided below for your reference:

Line 857-862: “Further, in addition to workplace interventions, policy changes on a national level, such as in-creasing federal minimum wage to a livable wage, will likely contribute to better mental health and reduce SPD across sociodemographic groups. Finally, shifting cultural norms in the United States (e.g., implementing workplace flexibilities that enhance autonomy; incentives and accountability measures) may positively impact health in the overall population and in terms of equity.”  

  1. Additionally, I suggest that a separate section should be devoted to explaining your research recommendations for future studies.

RESPONSE: Thank you for this suggestion. We have discussed recommendations for future research in a separate paragraph. We have also added text to further detail these recommendations. The text is provided below for your reference:

Line 810-838: “Future research studies should continue to assess the impact of occupational characteristics on mental health in these and other racial/ethnic (e.g., Native Americans), gender identity groups (e.g., nonbinary individuals), across sexual orientation groups, and across various types and levels of ability. Additionally, a focus on longitudinal and qualitative research is warranted to further unpack the stressors associated with working in various industries, occupational classes, and work environments, especially among populations with multiple minoritized identities in certain workplaces (e.g., Black women). Further, the interplay between the workplace and home environment as it relates to mental health should be further examined in groups that may be particularly burdened by their home environment (e.g., single parents, caregivers). Finally, research should continue to evaluate the implementation and effectiveness of various workplace mental health interventions, overall and within specific employment industries and occupational classes.”

  1. Finally, I really cannot see any relationship between the findings of your study and the aims thereof in terms of public health. In fact, in all subchapters of the manuscript, the term “public health” is not once mentioned. Therefore, I encourage you to review the content of your findings to favor an approach toward public health.

RESPONSE: Thank you for pointing this out to us. While most of the public health implications were implicit, we have expanded the text throughout the discussion to add language that explicitly discusses public health relevance and implications. The text is provided below for your reference:

Line 641-644: “This observation likely contributes to compromised mental health for structurally-marginalized groups, which highlights a need to address adverse working conditions to alleviate public health burden.”

Line 668-672: “Although this estimate should be interpreted with caution due to the large confidence interval as a result of a small sample size, this finding demonstrates the importance of continuing to conduct occupational health and public health research in general at the intersections of social identities (e.g., race/ethnicity-gender).”

Line 704-707: “Future research studies should consider how occupation-related factors may be impacted by and contribute to stress in the home environment and implement/evaluate solutions that may improve public health (e.g., respite services), especially for middle-aged adults.”

Line 718-720: “Increasing wages – at least to a livable wage – across industries can help reduce this public health burden and progress towards overall health equity.”

Line 782-784: “Additionally, these pathways may differ by sociodemographic groups, which is important to consider when creating equitable public health solutions.”

Reviewer 3 Report

The authors should be congratulated for what I regard as a near flawless manuscript. I recommend publication after addressing these minor issues:

15 – According to APA format, ‘the US’ should always be written “the United States”. Suggest you change this throughout the manuscript.

85 – ‘adults with poor health, which may include those with disabilities and chronic illnesses’. The word ‘those’ is othering. Poor health and chronic illnesses are also the same. Suggested edit: “adults with disabilities and chronic illnesses…”

367 – “vs.” should be “versus”. This recurs elsewhere in the manuscript.

368, 371 – “2-fold” should be “twofold”. The same occurred elsewhere (e.g., 437 “5-fold higher”).

373 – should be “… are dichotomized as younger or older than fifty-years old …”

520 – Start new paragraph at  “Our”.

552 – This sentence is too long. Split it into two.

606 – Insert subheading ‘5. Limitations of the study’.

Author Response

Reviewer #3: The authors should be congratulated for what I regard as a near flawless manuscript. I recommend publication after addressing these minor issues:

  1. 15 – According to APA format, ‘the US’ should always be written “the United States”. Suggest you change this throughout the manuscript.

RESPONSE: Thank you for this suggestion. We have updated the text throughout the manuscript to now read ‘United States’.

  1. 85 – ‘adults with poor health, which may include those with disabilities and chronic illnesses’. The word ‘those’ is othering. Poor health and chronic illnesses are also the same. Suggested edit: “adults with disabilities and chronic illnesses…”

RESPONSE: Thank you for this suggestion. This text was intended to describe the potential modifier (i.e., self-rated health status) we used. The survey question for this commonly-used variable asks participants if their health is in general ‘excellent, very good, good, fair, or poor’; therefore, we cannot explicitly determine if this variable only includes people with disabilities and chronic illnesses. However, we have updated the text for clarity and to refrain from othering certain groups. The updated text is provided below for your reference:

Line 105-109: “Finally, adults with self-reported poor health, which may include people with disabilities and chronic illnesses, may be limited in employment opportunities due unaccommodating and discriminating work environments (i.e., ableism), and thus may be additionally burdened by their occupational class and industry of employment [20] as well as SPD.

  1. 367 – “vs.” should be “versus”. This recurs elsewhere in the manuscript.

RESPONSE: We have updated the text throughout the manuscript to read ‘versus’.

  1. 368, 371 – “2-fold” should be “twofold”. The same occurred elsewhere (e.g., 437 “5-fold higher”).

RESPONSE: We have updated the text throughout the manuscript to read ‘twofold’, ‘fivefold’, and ‘sixfold.’

  1. 373 – should be “… are dichotomized as younger or older than fifty-years old …”

RESPONSE: We have updated the text with accurate information, which is provided below for your reference:

Line 471-473: “Age-race/ethnicity-stratified associations for occupational class, industry of employment and SPD are dichotomized as ‘younger than 50 years’ or ‘at least 50 years or older’ (Table S3).”

  1. 520 – Start new paragraph at “Our”.

RESPONSE: Thank you for this suggestion. As suggested, we have started a new paragraph. Below is the first sentence of the new paragraph for your reference:

Line 660: “There were also interesting findings among men.”

  1. 552 – This sentence is too long. Split it into two.

RESPONSE: Thank you for this suggestion. We have shortened the first sentence and divided the next sentence into two to make it easier to read. The original and updated text are both provided below for your reference:

Original: “As mentioned before, this may highlight the potential mental health impact of wage gap issues within this industry where there is an extremely large pay gap between the highest and lowest paid employees [24]. We also found that among NH-Black participants with lower household incomes, working in support services and laborer positions compared to management positions was associated with higher SPD, both overall and in several industries of employment (e.g., education services).”

Line 731-738: “As previously mentioned, this may highlight the potential mental health impact of wage gap issues within this industry including a large pay gap between the highest and lowest paid employees [24]. [24]. Increasing wages – at least to a livable wage – across industries can help reduce this public health burden and progress towards overall health equity. We also found that among NH-Black participants with lower household incomes, working in support services and laborer positions compared to management positions was associated with higher SPD. This observation was apparent in the overall sample and in several industries of employment (e.g., education services).”

  1. 606 – Insert subheading ‘5. Limitations of the study’.

RESPONSE: Thank you for this suggestion. We have added a limitations and strengths subheading.